

# Laminarin protects against hydrogen peroxide-induced oxidative damage in MRC-5 cells possibly via regulating NRF2

Xue Liu[1,2,*], Huaman Liu[3,*], Yi Zhai[4], Yan Li[5], Xue Zhu[3] and Wei Zhang[3]

[1] College of Traditional Chinese Medicine, Shandong University of Traditional Chinese Medicine, Jinan, China
[2] Department of Respiration, Shandong Provincial Chest Hospital, Jinan, China
[3] Department of Respiration, Affiliated Hospital of Shandong University of Traditional Chinese Medicine, Jinan, China
[4] Medical Department, Affiliated Hospital of Shandong University of Traditional Chinese Medicine, Jinan, China
[5] Department of Nursing, Zibo Central Hospital, Zibo, China
[*] These authors contributed equally to this work.

## ABSTRACT

Oxidative damage is a major cause of lung diseases, including pulmonary fibrosis. Laminarin is a kind of polysaccharide extracted from brown algae and plays vital roles in various biological processes. However, the functions and mechanisms of laminarin in pulmonary oxidative damage are poorly understood. This study aimed at investigating the protective effect of laminarin against pulmonary oxidative damage and underlying mechanisms. Human lung fibroblasts MRC-5 cells were treated with hydrogen peroxide to induce oxidative damage. Laminarin treatment was performed before or after hydrogen peroxide treatment, and then major indexes of oxidative damage, including superoxide dismutase (SOD), malondialdehyde (MDA), reduced glutathione (GSH) and catalase (CAT), were quantified by biochemical assays. The expression of oxidation-related factor, nuclear factor erythroid 2 like 2 (NRF2) was analyzed by qPCR, Western blot and immunofluorescence assay. NRF2 knockdown and overexpression were performed by cell transfection to reveal possible mechanisms. Results showed that laminarin treatment of 0.020 mg/mL for 24 h, especially the pre-treatment, could significantly relieve changes in SOD, MDA, GSH and CAT that were altered by hydrogen peroxide, and promote NRF2 mRNA ($P < 0.001$). NRF2 protein was also elevated by laminarin, and nuclear translocation was observed. Factors in NRF2 signaling pathways, including KEAP1, NQO1, GCLC and HO1, were all regulated by laminarin. Roles of NRF2 were tested, suggesting that NRF2 regulated the concentration of SOD, MDA, GSH and CAT, suppressed KEAP1, and promoted NQO1, GCLC and HO1. These findings suggested the protective role of laminarin against pulmonary oxidative damage, which might involve the regulation of NRF2 signaling pathways. This study provided information for the clinical application of laminarin to pulmonary diseases like pulmonary fibrosis.

Corresponding author
Wei Zhang, huxizhijia@126.com

## INTRODUCTION

As one of the vital respiratory organs of human, lungs play important roles in gas exchange as well as excretion, fluid exchange, regulation of acid–base balance and other biological processes. Nevertheless, lung tissue is exposed to adverse impacts from the external environment, which call for a higher capacity of lungs to fight against the oxidative damage. Oxides in lungs mainly derive from systemic circulation, gas inhalation and the pathological processes of the body. For example, the enhancive oxygen consumption during respiratory burst induces the production of several oxides or enzyme like myeloperoxidase, hypochlorous acid and hydrogen peroxide, which oxidize sulfhydryl groups and cause cellular membrane damage (*Goraca & Józefowicz-Okonkwo, 2007*). Ozone inhalation may lead to inflammation and pulmonary oxidative damage, mediating the activity of factors such as toll-like receptor 4 (*Connor, Laskin & Laskin, 2012*; *Li et al., 2011*). Moreover, cigarette smoke activates the production of nitric oxide, further aggravating pulmonary oxidative damage (*Ren et al., 2000*; *Turanlahti et al., 2000*).

Pulmonary oxidative damage may cause severe diseases, such as chronic obstructive pulmonary disease, pulmonary fibrosis, respiratory distress syndrome and even lung cancer (*Fernandez-Bustamante & Repine, 2014*; *Sunnetcioglu et al., 2016*). Clinical trials have indicated the positive correlation between pulmonary fibrosis and the level of oxidative stress. Levels of antioxidants, including reduced glutathione (GSH), super dismutase (SOD) and catalase (CAT), were markedly reduced in the lung of IPF patients (*Gao et al., 2007*). Besides, ROS levels were significantly up-regulated in bleomycin-induced pulmonary fibrosis rats (*Teixeira et al., 2008*). Extensive efforts have been made to discover substances relieving pulmonary oxidative damage. For instance, hydrogen therapy was found possessing the potential of reducing irradiation-induced pulmonary oxidative stress (*Terasaki et al., 2011*). Angiotensin-(1–7) and nitric oxide synthase inhibitor L-NAME showed inhibitive effects on pulmonary oxidative damage in rodents (*De Bittencourt Pasquali et al., 2012*; *Lu et al., 2016*). However, there remains an urgent need for nontoxic and antioxidant therapies to prevent and remedy pulmonary oxidative damage and pulmonary fibrosis.

Laminarin is a $\beta(1-3)$polysaccharide extracted from brown algae. With the development of the purification technique, laminarin is showing its application prospects in clinical trials. Some studies have found the application value of laminarin as dietary supplement since laminarin altered gut microbiota and had hepatoprotective effects (*Neyrinck, Mouson & Delzenne, 2007*; *Nguyen et al., 2016*). Moreover, the anti-cancer function of laminarin has been reported, which indicated that laminarin could induce cancer cell apoptosis and suppress angiogenesis (*Hoffman et al., 1996*; *Ji & Ji, 2014*). Further, it has been reported that laminarin acted as an antioxidant (*Zhou et al., 2009*), although the mechanism of this effect was not understood. Thus, more information is still necessary to reveal the role of laminarin in pulmonary oxidative damage.

The aim of this study is to investigate the function of laminarin in hydrogen peroxide-induced pulmonary oxidative damage by NRF2/ARE signaling pathway. Human lung fibroblasts MRC-5 cells were treated with hydrogen peroxide to induce oxidative damage.

Laminarin pre-treatment and after treatment were performed, after which major indexes of oxidative damage, including SOD, malondialdehyde (MDA), GSH and CAT, were quantified by biochemical assays. We also tried to elucidate the regulatory mechanism of laminarin in pulmonary oxidative damage through detecting expression changes of nuclear factor erythroid 2 like 2 (NRF2) and key factors in NRF2 signaling pathways. These findings would help to understand the role and functional mechanism of laminarin in controlling pulmonary oxidative damage and diseases such as pulmonary fibrosis.

## MATERIALS AND METHODS

### The detection of laminarin antioxidant activity

The antioxidant activity of laminarin (Sigma-Aldrich, Saint Louis MO, USA) was detected by clearance rate of hydroxyl free radical ($\cdot$OH) and superoxide radical ($O_2^-\cdot$). Vitamin C (Vc) was used for positive control, and ddH$_2$O$_2$ was used for negative control. The clearance rate of hydroxyl free radical and superoxide radical were detected by Smirnoff's method and Ponti's method respectively (*Smirnoff & Cumbes, 1989*; *Ponti et al., 1978*). Antioxidants were added to the reaction systems with different concentrations (0, 1, 2, 4, 8 and 16 mM; 0, 0.01, 0.02, 0.04, 0.08, and 0.16 mM). Optical density (OD) values at 510 nm and 560 nm were measured by microplate reader Multiskan Go (Thermo Scientific, USA). The calculation of clearance rate was as follow: Clearance rate = [(OD of blank control) – (OD of antioxidants)]/(OD of blank control) $\times$ 100%.

### Cell culture

Human lung fibroblast cells MRC-5 (American Type Culture Collection, Manassas VA, USA) were cultured in Dulbecco's modified Eagle's medium supplemented with 10% fetal bovine serum (Gibco, Carlsbad CA, USA). Cells were incubated in humidified atmosphere containing 5% CO$_2$ at 37 °C and passaged when the confluence reached about 80%.

### Cell transfection

Cell transfection was performed to knock down and overexpress NRF2. For NRF2 knockdown, MRC-5 cells in the exponential phase were transfected with the specific small interfering RNA (siRNA) designed for *NRF2* gene or the negative control siRNA (RiboBio, Guangzhou, China) using Lipofectamine 2000 (Invitrogen, Carlsbad CA, USA) according to the manufacturer's instruction. For NRF2 overexpression, the complete coding sequence of *NRF2* mRNA was ligated to pcDNA3.1 overexpression vector (Thermo Scientific, Carlsbad CA, USA) and the correct clone was screened by sequencing. Then the NRF2 recombined vector (pcDNA3.1-NRF2) was transfected to MRC-5 cells in the exponential phase using Lipofectamine 2000, and the blank vector (pcDNA3.1) was transfected as a negative control. Levels of NRF2 mRNA and protein were detected at 48 h post transfection.

### Hydrogen peroxide and laminarin treatment

Oxidative damage in MRC-5 cells was induced by hydrogen peroxide treatment. Hydrogen peroxide (HepengBio, Shanghai, China) was added to the culture medium of cells with
**Table 1  Primers used in real-time quantitative PCR (qPCR).**

| Primer | Sequence (5′–3′) | Product size (bp) |
|---|---|---|
| NRF2 | Forward: TCCGGGTGTGTTTGTTCCAA<br>Reverse: CGCCCGCGAGATAAAGAGTT | 88 |
| KEAP1 | Forward: GTCCCCTACAGCCAAGGTCC<br>Reverse: ACTCAGTGGAGGCGTACATC | 175 |
| NQO1 | Forward: GGTTTGGAGTCCCTGCCATT<br>Reverse: ACCAGTGGTGATGGAAAGCA | 134 |
| GCLC | Forward: GAGGTCAAACCCAACCCAGT<br>Reverse: AAGGTACTGAAGCGAGGGTG | 92 |
| HO1 | Forward: TCCTGGCTCAGCCTCAAATG<br>Reverse: CGTTAAACACCTCCCTCCCC | 108 |
| GAPDH | Forward: ATCTTCTTTTGCGTCGCCA<br>Reverse: TTAAAAGCAGCCCTGGTGACC | 202 |

different concentrations (0, 200, 400, 600, 800 and 1,000 μM) and then the cells were incubated for 6, 12 or 24 h. Laminarin was added to the medium to different concentrations at 1 h before or after the addition of hydrogen peroxide, and after 24 h of incubation, 3-[4,5-dimethylthiazol-2-yl]-2,5-diphenyl tetrazolium bromide (MTT) assay was performed to determine the proper concentration and time of incubation. Cell transfection was performed at 48 h before the cells were treated by hydrogen peroxide or laminarin.

## Real-time quantitative PCR (qPCR)

Total RNA of MRC-5 cells was extracted using TRIzol (Invitrogen, Carlsbad, CA, USA) and purified with RNA Purification Kit (Tiangen, Beijing, China). The quality and quantity of RNA was examined by agarose gel electrophoresis and NanoDrop 2000 (Thermo Scientific, Waltham, MA, USA). The complementary DNA (cDNA) was synthesized by ReverAid First Strand cDNA Synthesis Kit (Thermo Scientific, Waltham, MA, USA) from 1 μg RNA for each sample. The expression level of *NRF2*, *KEAP1*, *NQO1*, *GCLC* and *HO1* was quantified by qPCR, which was conducted on LightCycler 480 (Roche, Basel, Switzerland). Each PCR system contained cDNA (20 ng) and specific primers (Table 1) for the target mRNA, and the reaction was catalyzed by SYBR Green I Master (Roche, Basel, Switzerland) according to the manufacturer's instruction. The experiment was performed in triplicate. Data were analyzed by $2^{-\Delta\Delta Ct}$ method with *GAPDH* as a reference gene.

## Western blot

Total protein, nucleoprotein and cytoplasmic protein of MRC-5 cells were extracted using Radio Immunoprecipitation Assay Buffer (Beyotime, Shanghai, China) and EpiQuik Nuclear Extraction Kit (Epigentek, Farmingdale NY, USA), respectively. Protein samples were quantified by Bio-Rad Protein Assay (Bio-Rad, Hercules CA, USA) and then loaded to sodium dodecyl sulfate-polyacrylamide gel electrophoresis. Protein bands on the gel were then transferred to polyvinylidene fluoride membranes, which were then blocked in 5% skim milk for 4 h at room temperature. The membranes were incubated in Tris-buffered saline and Tween 20 (TBST) containing the specific primary antibodies against

NRF2, KEAP1, NQO1, GCLC, HO1 and GAPDH (ab89443, ab150654, ab28947, ab55435, ab13248 and ab8245; Abcam, Cambridge, UK) overnight at 4 °C. After washed in TBST for five times, the membranes were incubated in horse radish peroxidase-conjugated secondary antibodies (ab6708, Abcam) for 2 h at room temperature, and then washed again in TBST for five times. Protein signals were developed using ECL Plus Western Blotting Substrate (Pierce, Carlsbad CA, USA).

## MTT assay

MTT assay was performed to assess viability and inhibition rate of MRC-5 cells post treatment. The cells ($5 \times 10^4$/mL) were added to 96-well plates and incubated at 37 °C for 24 h. Then, the culture medium was replaced by medium containing hydrogen peroxide or laminarin for further incubation. The assay was performed at different time points using MTT Cell Proliferation and Cytotoxicity Assay Kit (Beyotime) according to the manufacturer's instruction. OD at 490 nm was measured by a microplate reader Multiskan Go. Biological experiments were repeated six times. Inhibition rate was calculated as [(OD of untreated group) − (OD of treated group)]/(OD of untreated group) × 100%.

## Immunofluorescence

NRF2 protein expression and location in MRC-5 cells were detected by immunofluorescence. Cell smears were fixed in 4% paraformaldehyde overnight and then washed in phosphate-buffered saline (PBS) for three times. The smears were incubated in hydrogen peroxide (3%) in methyl alcohol for 10 min, goat serum (Beyotime, Shanghai, China) for 20 min and the primary antibody for NRF2 for 2 h at 37 °C in sequence, with three times of wash in PBS between steps. Then fluorescein isothiocyanate (FITC)-conjugated secondary antibody (ab7064; Abcam, Cambridge, UK) was added to the smears, which were incubated in the dark for 1 h at 37 °C. After washed in PBS for three times, the smears were counterstained with 4′,6-diamidino-2-phenylindole (DAPI; Beyotime, Shanghai, China) in the dark at room temperature for 5 min. The smears were mounted and observed under a fluorescence microscope (Olympus, Tokyo, Japan).

## Biochemical assays

Cellular SOD, MDA, GSH and CAT concentrations were detected by biochemical methods using commercial kits: SOD Activity Assay Kit (BioVision, Milpitas CA, USA), MDA Detection Kit (Solarbio, Shanghai, China), GSH and GSSG Assay Kit (Beyotime) and Human CAT ELISA Kit (Uscn Life Science Inc., Wuhan, China). Experiments were conducted according the manufacturers' instruction, and detection was performed by a microplate reader (Thermo Scientific, Waltham, MA, USA).

## Statistical analysis

All the experiments were performed at least in triplicate and results were expressed as mean ± standard deviation. Data were analyzed by Student's $t$ test and one-way analysis of variance using SPSS 20 (New York NY, USA). Differences with $P < 0.05$ were considered to be statistically significant.

## RESULTS

### The antioxidant activity of laminarin

Laminarin was purified from kelp, and it could resist oxidative damage. In order to assess its antioxidant activity, the clearance rate of hydroxyl free radical and superoxide radical was detected. The results shown that the clearance rates were increased following the raising of the concentration (Fig. S1). It was found that laminarin possessed strong antioxidant activity against hydroxyl free radical, and it had the similar consequence with Vc (Fig. S1A). Meanwhile, laminarin also had the ability of scavenging superoxide radical (Fig. S1B). According to these results, it could be indicate that laminarin was a good antioxidant, which could clear free radicals effectively.

### Laminarin relieves oxidative damage caused by hydrogen peroxide in MRC-5 cells

The therapeutic effects of laminarin were reported in various diseases. In order to assess the functions of laminarin in oxidative damage of human lung fibroblast cells, this study first determined the proper concentration and time of incubation by analyzing cell viability and inhibition rate after MRC-5 cells were treated by hydrogen peroxide or laminarin of different concentrations and different times of incubation. Results showed that a hydrogen peroxide concentration of over $600 \, \mu M$ induced severe inhibition on MRC-5 cell viability when detected by MTT at 6, 12 and 24 h post treatment ($P < 0.01$, Fig. 1A), and all the five concentrations suppressed cell viability when the detection was performed at 24 h post treatment ($P < 0.05$). The inhibition rate calculated on the basis of these OD values was raised with the increased concentration and incubation time (Fig. 1B). Accordingly, a hydrogen peroxide concentration of $600 \, \mu M$ and treatment for 24 h were set in the following experiments. After treated with laminarin of different concentrations (Grade 0 to 11) for 24 h, cells were applied to MTT assay, which showed that cell viability was decreased with the increased laminarin concentration, and that concentration of more than 0.0390625 mg/mL significantly impaired cell viability ($P < 0.05$, Fig. 1C). Also, the inhibition rate was increased with the ascending laminarin concentration (Fig. 1D). In order to guarantee cell viability, we chose 0.1953125 mg/mL as the proper laminarin concentration in the following experiments.

SOD, MDA, GSH and CAT are major indexes that are altered during oxidative damage. In this study, the four indexes were detected after hydrogen peroxide and laminarin treatment. Results showed that SOD, GSH and CAT levels in MRC-5 cells were decreased by hydrogen peroxide (Figs. 2A, 2C and 2D), while MDA level was increased (Fig. 2B). Laminarin caused diverse changes that SOD, GSH and CAT were up-regulated and MDA down-regulated. Laminarin pre-treatment relieved the change caused by hydrogen peroxide, but the effect of laminarin after treatment was weaker. Moreover, level changes of the four substances got more obvious with the increasing incubation time. It could be implied from these data that laminarin treatment, especially the pre-treatment, might help to attenuate oxidative damage caused by hydrogen peroxide in MRC-5 cells.

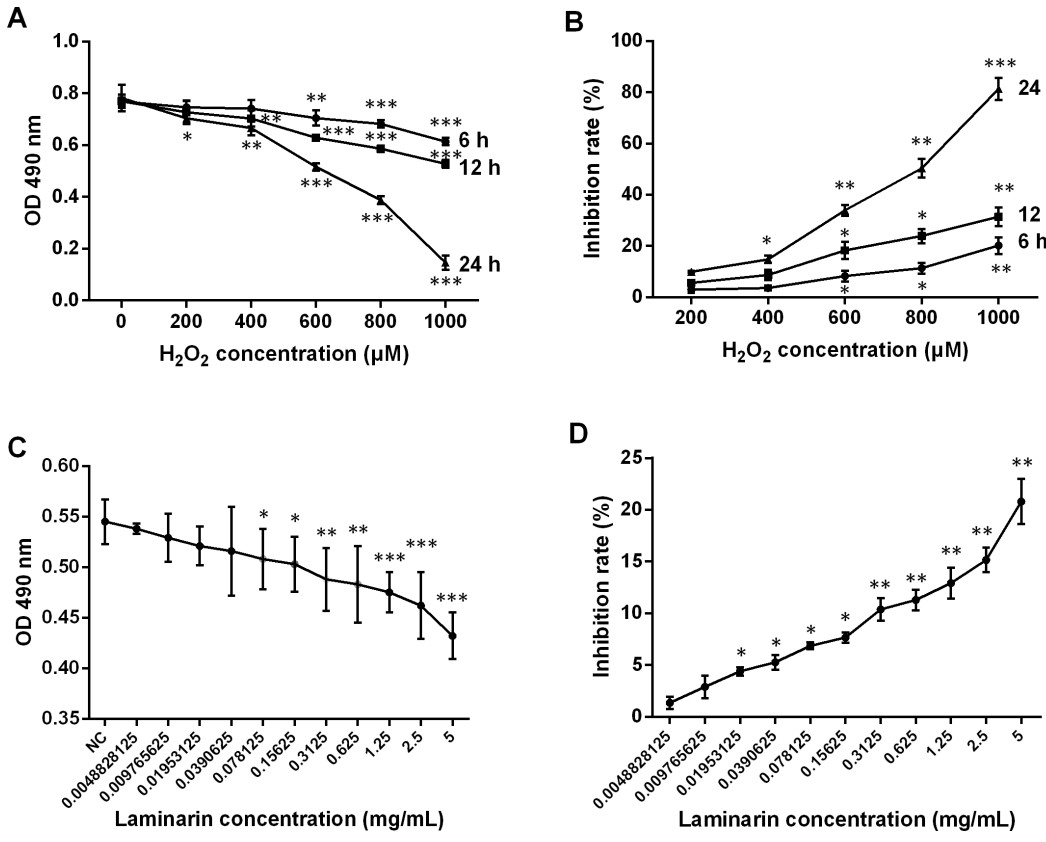

**Figure 1** Change of viability and inhibition rate of human lung fibroblast cells MRC-5 by hydrogen peroxide ($H_2O_2$) and laminarin of different concentrations and incubation time. (A) MRC-5 cells were treated with $H_2O_2$ (0, 200, 400, 600, 800 or 1,000 µM) for 6, 12 or 24 h, after which optical density (OD) at 490 nm was measured in MTT assay to assess cell viability. (B) Inhibition rate by $H_2O_2$ calculated according to MTT results. (C) MRC-5 cells were treated with laminarin of different concentrations (grade 0–11) for 24 h, after which MTT assay was performed to measure OD 490 nm. (D) Inhibition rate by laminarin calculated according to MTT results. * $P < 0.05$, ** $P < 0.01$ and *** $P < 0.001$ compared with concentration Grade 1.

## NRF2 is involved in the regulation of oxidative damage by laminarin

Existed research has found evidence that NRF2 may participate in the regulation of oxidative damage, hence we hypothesized that NRF2 was also involved in the regulation of oxidative damage by laminarin in MRC-5 cells. To test the hypothesis, we first detected NRF2 expression in MRC-5 cells after hydrogen peroxide and laminarin treatment. qPCR results showed that no significant change was detected at 1 or 2 h post treatment. However, *NRF2* mRNA was significantly down-regulated by hydrogen peroxide at 6, 12 and 24 h post treatment compared to Group control ($P < 0.01$, Fig. 3A), and laminarin significantly increased *NRF2* mRNA level at the three time points ($P < 0.05$). Laminarin pre-treatment obviously relieved the down-regulation of *NRF2* mRNA by hydrogen peroxide ($P < 0.01$), but the after treatment only had significant effect when detection was conducted at 24 h post treatment ($P < 0.05$).

NRF2 protein level was also examined. Western blot showed that NRF2 was expressed mainly in cytoplasm (Fig. 3B). The cellular level of NRF2 was suppressed by hydrogen

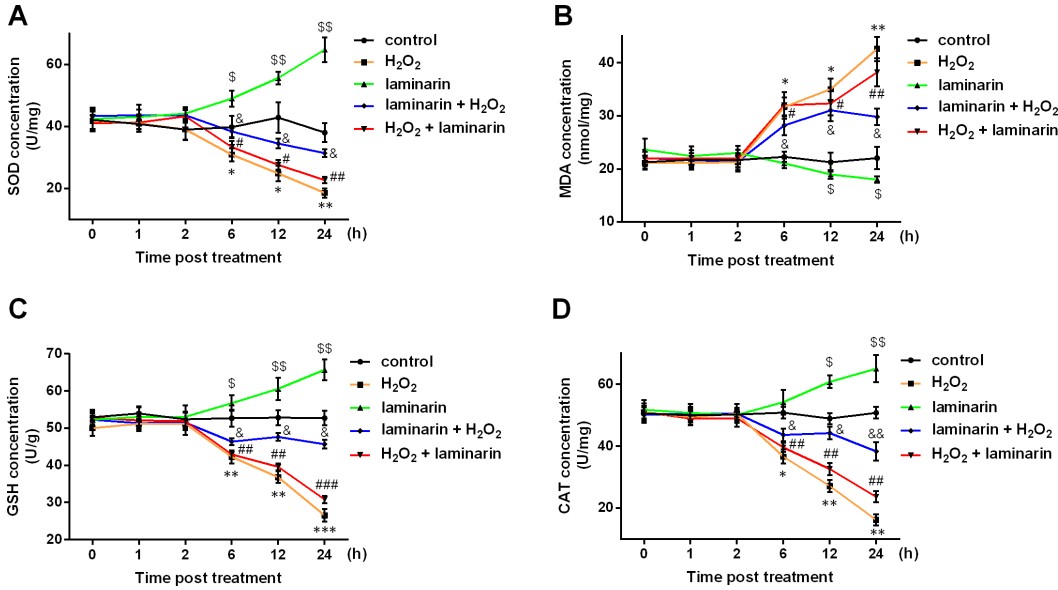

**Figure 2  Laminarin relieves the level change of superoxide dismutase (SOD), malondialdehyde (MDA), reduced glutathione (GSH) and catalase (CAT) caused by hydrogen peroxide ($H_2O_2$) in MRC-5 cells.** MRC-5 cells were treated with $H_2O_2$ (600 $\mu$M) or laminarin (0.020 mg/mL), and biochemical detection for SOD (A), MDA (B), GSH (C) and CAT (D) was performed at 0, 1, 2, 6, 12 and 24 h post treatment. ** $P < 0.01$ and *** $P < 0.001$ in comparison between Group control and Group $H_2O_2$. # $P < 0.05$ in comparison between Group $H_2O_2$ and Group $H_2O_2$ + laminarin. \$ $P < 0.05$, \$\$ $P < 0.01$ and \$\$\$ $P < 0.001$ in comparison between Group control and Group laminarin. && $P < 0.01$ and &&& $P < 0.001$ in comparison between Group $H_2O_2$ and Group laminarin + $H_2O_2$.

peroxide and induced by laminarin. As for the latter, cells pre-treated with laminarin induced NRF2 up-regulation in the nucleus compared to those only treated with hydrogen peroxide, while laminarin after treatment mainly caused the promotion of NRF2 in cytoplasm. The location change of NRF2 protein was further detected by immunofluorescence (Fig. 3C), which indicated that the pre-treatment of laminarin could induce NRF2 expression in nucleus in hydrogen peroxide-induced cells, and laminarin after treatment could increase cytoplasmic expression of NRF2. Anyway, laminarin could attenuate the down-regulation of NRF2 caused by hydrogen peroxide, but the translocation of NRF2 and the different effects of laminarin pre-treatment and after treatment required further investigation.

Studies have discovered several factors in NRF2 signaling pathways, including kelch-like ECH associated protein 1 (KEAP1), NAD(P)H quinone dehydrogenase 1 (NQO1), glutamate-cysteine ligase catalytic subunit (GCLC) and heme oxygenase 1 (HO1). Hence the mRNA level of the four factors were also quantified by qPCR in this study. Results showed that *KEAP1* mRNA was up-regulated by hydrogen peroxide after 6, 12 and 24 h of treatment ($P < 0.05$, Fig. 4A). Laminarin pre-treatment significantly relieved the effect of hydrogen peroxide on *KEAP1* mRNA at 6, 12 and 24 h post treatment ($P < 0.05$). However, laminarin after treatment did not show significant mitigatory effects when the cells were treated for less than 24 h. Also, the effect of laminarin treatment alone was bewildering, which deserved more detailed research. For *NQO1*, *GCLC* and *HO1* mRNA levels, similar

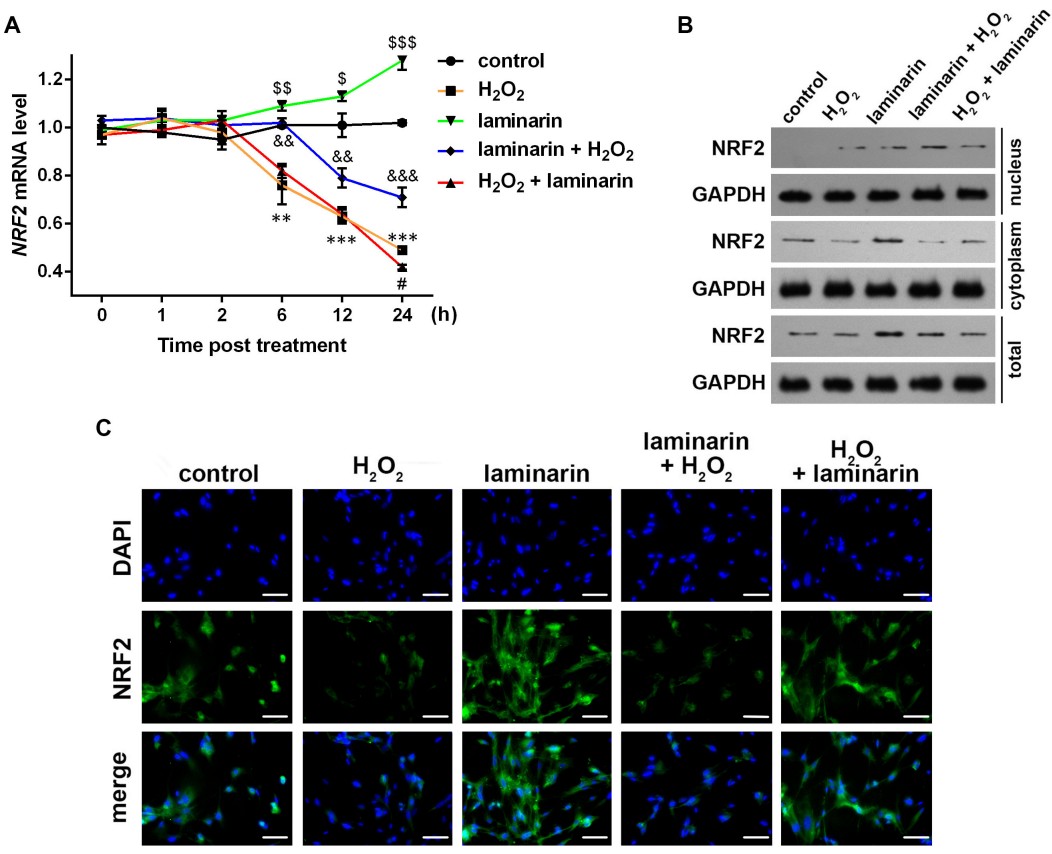

**Figure 3** **Expression change of nuclear factor erythroid 2 like 2 (NRF2) induced by hydrogen per-oxide (H$_2$O$_2$) and laminarin in MRC-5 cells.** MRC-5 cells were treated with H$_2$O$_2$ (600 μM) and lami-narin (0.020 mg/mL). (A) NRF2 mRNA level quantified by qPCR at 0, 1, 2, 6, 12 and 24 h post treatment. ** $P < 0.01$ and *** $P < 0.001$ in comparison between Group control and Group H$_2$O$_2$. # $P < 0.05$ in comparison between Group H$_2$O$_2$ and Group H$_2$O$_2$ + laminarin. $P < 0.05$, $P < 0.01$ and $P < 0.001$ in comparison between Group control and Group laminarin. && $P < 0.01$ and &&& $P < 0.001$ in com-parison between Group H$_2$O$_2$ and Group laminarin + H$_2$O$_2$. (B) NRF2 protein level detected by Western blot at 24 h post treatment. GAPDH was used as an internal reference. (C) NRF2 protein expression and located detected by immunofluorescence at 24 h post treatment. NRF2 protein (green) was visualized by fluorescein isothiocyanate-conjugated antibodies, and 4′,6-diamidino-2-phenylindole (DAPI) was used to dye the nucleus (blue). Bar indicates 10 μm.

changing patterns were observed at 6, 12 and 24 h post treatment (Figs. 4B–4D): hydrogen peroxide suppressed ($P < 0.01$), and laminarin promoted mRNA levels of these factors ($P < 0.05$). Besides, laminarin pre-treatment greatly attenuated the suppressive effect of hydrogen peroxide on these mRNAs ($P < 0.01$), while laminarin after treatment did not show obvious consequences, possibly due to the limited time range of experiments. Based on these data, laminarin could induce the expression of NRF2 and the related factors KEAP1, NQO1, GCLC and HO1, which implied that NRF2 and its pathways might be involved in the mechanism of laminarin in MRC-5 cells.

To get more supporting information for our hypothesis, NRF2 was knocked down and overexpressed to determine whether it participated in the regulation of oxidative damage. After the cells were transfected with NRF2-specific siRNA, *NRF2* mRNA level was

 

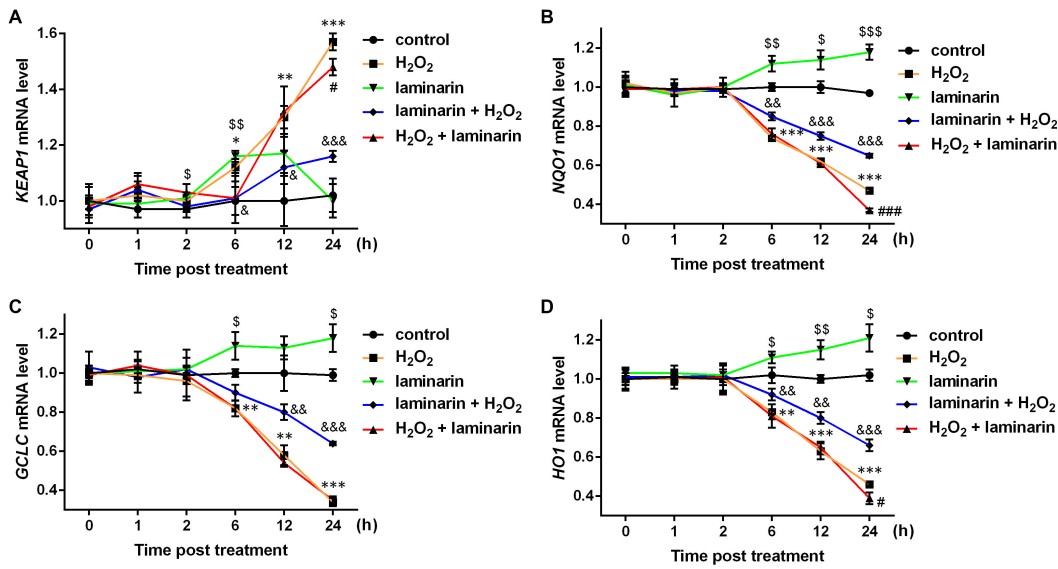

**Figure 4** **mRNA level change of KEAP1, NQO1, GCLC and HO1 by $H_2O_2$ and laminarin in MRC-5 cells.** Cells were treated with $H_2O_2$ (600 μM) and laminarin (0.020 mg/mL), after which mRNA level of KEAP1 (A), NQO1 (B), GCLC (C) and HO1 (D) were quantified by qPCR at 0, 1, 2, 6, 12 and 24 h post treatment. * $P < 0.05$, ** $P < 0.01$ and *** $P < 0.001$ in comparison between Group control and Group $H_2O_2$. # $P < 0.05$ and ### $P < 0.001$ in comparison between Group $H_2O_2$ and Group $H_2O_2$ + laminarin. $ $P < 0.05$, $$ $P < 0.01$ and $$$ $P < 0.001$ in comparison between Group control and Group laminarin. & $P < 0.05$, && $P < 0.01$ and &&& $P < 0.001$ in comparison between Group $H_2O_2$ and Group laminarin + $H_2O_2$.

significantly suppressed, as quantified by qPCR in MRC-5 cells untreated and treated by hydrogen peroxide or laminarin ($P < 0.05$, Fig. 5A). Besides, when NRF2 was knocked down, hydrogen peroxide suppressed and laminarin promoted *NRF2* mRNA level. NRF2 protein level in cells possessed similar changing patterns (Fig. 5B). Cellular concentration of SOD, MDA, GSH and CAT was also quantified after NRF2 was knocked down (Fig. 5C) and results showed that NRF2 knockdown could inhibit SOD, GSH and CAT levels and promote MDA level, which suggested the involvement of NRF2 in regulating oxidative damage in MRC-5 cells.

Expression changes in KEAP1, NQO1, GCLC and HO1 were analyzed when NRF2 was up-regulated by its overexpression vector pcDNA3.1-NRF2. qPCR results indicated that *NRF2* mRNA level was significantly up-regulated by pcDNA3.1-NRF2 in MRC-5 cells with or without laminarin treatment ($P < 0.05$, Fig. 6A), suggesting the effective NRF2 overexpression. In both laminarin-treated and untreated MRC-5 cells, NRF2 overexpression could suppress *KEAP1* mRNA and promote *NQO1*, *GCLC* and *HO1* mRNA levels ($P < 0.05$, Figs. 6B–6E). Besides, laminarin treatment further promoted the effect of NRF2 overexpression on these factors. Similar protein changing patterns of these factors were also observed in Western blot results (Fig. 6F). Collectively, NRF2 could regulate KEAP1, NQO1, GCLC and HO1. Together with the abovementioned results, laminarin might elevate NRF2 and related factors, thus attenuating the oxidative damage caused by hydrogen peroxide in MRC-5 cells.

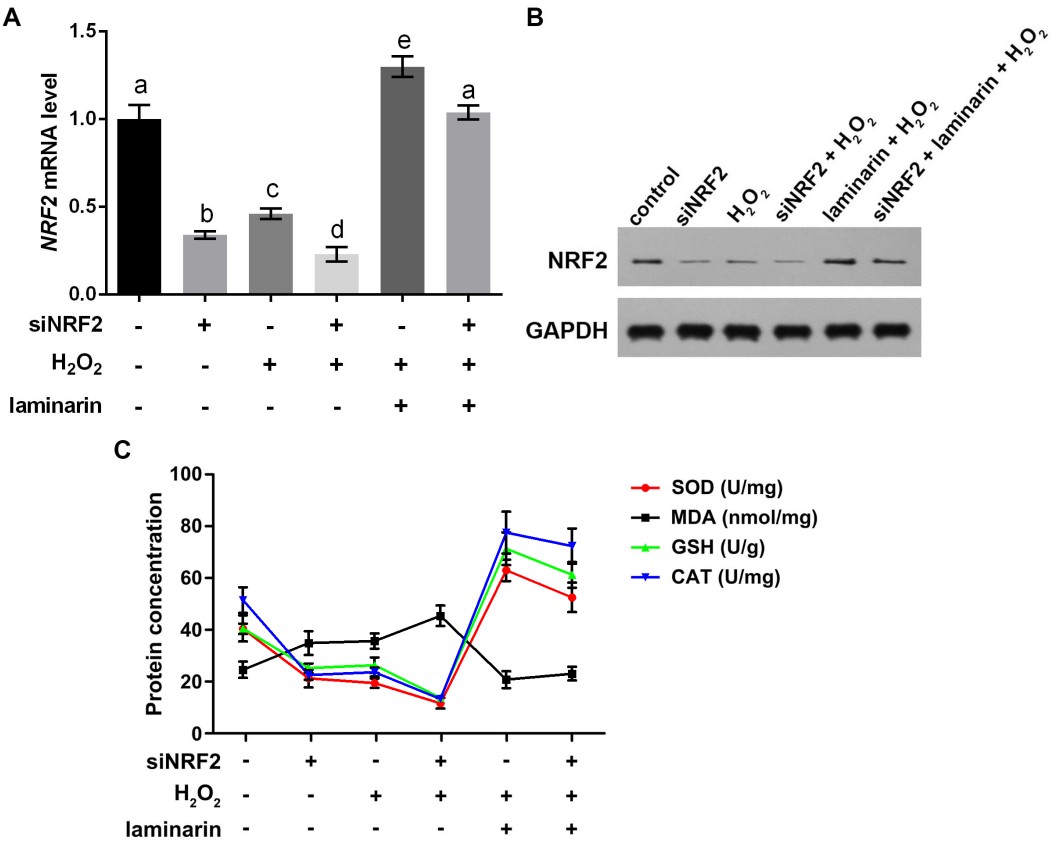

**Figure 5** **Nuclear factor erythroid 2 like 2 (NRF2) regulates oxidative damage in MRC-5 cells.** NRF2 was knocked down by its specific siRNA (siNRF2), and cells were treated with hydrogen peroxide ($H_2O_2$, 600 $\mu$M) and laminarin (0.020 mg/mL). Assays were performed at 24 h post treatment. (A) *NRF2* mRNA level quantified by qPCR. (B) NRF2 protein level detected by Western blot. GAPDH was used as an internal reference. (C) Cellular concentration of superoxide dismutase (SOD), malondialdehyde (MDA), deduced glutathione (GSH) and catalase (CAT) quantified by biochemical assays. Values with different letters indicate statistical significance ($P < 0.05$) .

## DISCUSSION

Laminarin has been reported to have protective effects against various diseases. Although existed studies have found the mitigatory role of laminarin in sepsis-induced pulmonary oxidative damage of rats (*Cheng et al., 2011*), little is known about the underlying mechanisms. This study investigated the role of laminarin in protecting MRC-5 cells against hydrogen peroxide-induced oxidative damage, as well as the possible mechanism regarding the regulation of NRF2.

SOD is a kind of antioxidants that catalyzes the conversion from superoxide radicals to hydrogen peroxide, participating in the protection against oxidant-related lung disorders (*Kinnula & Crapo, 2003*). Increased GSH prevents oxidative damage (*Jat et al., 2013*; *Weisel et al., 2006*), and decreased catalase may contribute to oxidative damage (*Escribano et al., 2015*). Besides, MDA has long been used as a biomarker for oxidative damage to lipids (*Lykkesfeldt, 2007*). In this study, we tested hydrogen peroxide of different concentration in

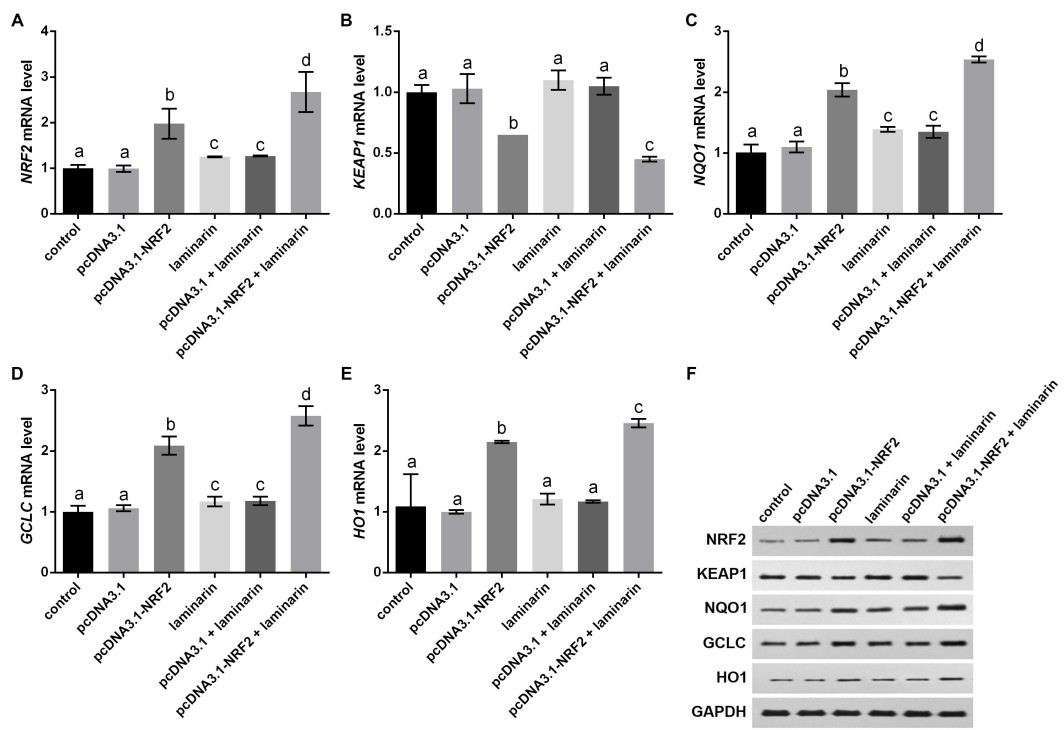

**Figure 6 NRF2 regulates KEAP1, NQO1, GCLC and HO1 in MRC-5 cells treated with laminarin.**
MRC-5 cells were transfected with the overexpression vector of NRF2 (pcDNA3.1-NRF2) or blank vector (pcDNA3.1) as a control, and then treated with laminarin (0.020 mg/mL). qPCR was performed to quantify the mRNA level of NRF2 (A), KEAP1 (B), NQO1 (C), GCLC (D) and HO1 (E) at 24 h post laminarin treatment. (F) Protein level of factors detected by Western blot. GAPDH was used as an internal control. Values with different letters indicate statistical significance ($P < 0.05$).

MRC-5 cells, and found concentration over 600 μM led to significant suppression on cell viability and high inhibition rate at all the time points examined. After being treated with hydrogen peroxide (600 μM) for 24 h, MRC-5 cells exhibited significantly lower SOD, GSH and CAT levels and significant higher MDA level. Together with the previously reported function of SOD, MDA, GSH and CAT in oxidative damage, data in this study suggested that hydrogen peroxide successfully induced oxidative damage in MRC-5 cells.

Laminarin treatment in MRC-5 cells had obvious influences of increasing SOD, GSH and CAT levels, and meanwhile suppressing MDA level, implying the protective function of laminarin against oxidative damage. Similar results have been reported in rats of exhaustive exercise, where laminarin reduced oxidative stress as indicated by elevated SOD, GSH and CAT levels and decreased MDA level (*Cheng, Liang & Li, 2012*). The effects of laminarin pre-treatment and after treatment were compared, and results showed that pre-treatment had more significant effects on all the four indexes, implying that laminarin might play preventive roles against pulmonary oxidative damage. The obvious effects of laminarin after treatment might be detected if the treatment lasted for more than 24 h. More investigation *in vivo* would help to verify the speculation.

Based on the above analysis, we further studied the functional mechanism of laminarin. NRF2 is a vital factor defending against chemical and oxidative stress caused by exogenous

substances. KEAP1 was reported as an NRF2 repressor that was up-regulated during oxidative stress, thus leading to the inhibition of NRF2 (*Kim & Vaziri, 2010*; *Kobayashi et al., 2006*). Results of this study indicated that hydrogen peroxide treatment promoted KEAP1 level and suppressed NRF2. Laminarin up-regulated NRF2 expression, but its influence on KEAP1 was debatable. Moreover, overexpression of NRF2 suppressed KEAP1 expression. One possible explanation of this phenomenon might lie in the mutual regulation between NRF2 and KEAP1. A growing body of literature has proved the existence of NRF2 regulation mechanisms other than the KEAP1-NRF2 signaling pathway (*Bryan et al., 2013*). Some researchers maintained that increasing NRF2 level would saturate the antioxidant response element (ARE) binding sites in KEAP1, compelling free NRF2 to translocate to nucleus and activate the transcription of its downstream genes (*Zhang, 2006*). This explanation was supported by some data of this study that laminarin pre-treatment increased the proportion of nuclear NRF2 protein, which further activate the expression of NQO1, GCLC and HO1. Nonetheless, it was possible that NRF2 was regulated by other factors that might also affect KEAP1 level.

The transcription of NQO1, GCLC and HO1 is regulated by various factors including NRF2. Existed research has indicated that induced NQO1 and HO1 levels were accompanied with the accumulation NRF2 in nucleus, thus protecting cells against oxidative damage (*Zhang et al., 2008*). In sepsis-induced acute lung injury, up-regulation of the NRF2-GCLC signaling pathway was associated with enhancement of GSH and reduction of oxidative damage (*Zong & Zhang, 2017*). Consistent with former research, this study found the elevated mRNA and protein levels of NQO1, GCLC and HO1 by NRF2 overexpression, suggesting that NRF2 could promote the expression of NRF2 when it stimulated the cell. Then, NRF2 could activate the expression of anti-oxidant enzymes in downstream by NRF2/ARE signaling pathway. Further, laminarin exerted anti-oxidation, which might further facilitate the remission of oxidative damage. The specific type of interaction should be explored further.

The influence of NRF2 on oxidative damage indexes was also examined in order to elucidate the involvement of NRF2 in laminarin functional mechanisms. Biochemical experiments indicated that knockdown of NRF2 could increase MDA concentration and decrease concentration of SOD, GSH and CAT, enhancing the effects of laminarin in MRC-5 cells. Numerous studies have proved the protective role of NRF2 against oxidation in various cell types, keratinocytes (*Madduma Hewage et al., 2016*; *Zheng et al., 2014*), pancreatic β-cells (*Dinić et al., 2016*), mesenchymal stem cells (*Loseva et al., 2012*), amongst others, elevating SOD, GSH and CAT cellular concentration. Besides, up-regulation of NRF2 was accompanied with suppressed MDA level during the amelioration of oxidative damage (*Xie et al., 2014*). Together with the findings of this study, NRF2 was a powerful alleviator of oxidative damage, and it was thus reasonable to deduce that the up-regulation of NRF2 might be one mechanism for elucidating why laminarin attenuated hydrogen peroxide-induced oxidative damage in MRC-5 cells.

Furthermore, NRF2 also participates in the regulation of oxidative damage that promotes pulmonary fibrosis. The NRF2/ARE signaling pathways activates the expression of many antioxidant enzymes and defense proteins which can subsequently fight against the

enhanced oxidative damage during pulmonary fibrosis (*Walters, Cho & Kleeberger, 2008*). Research in bleomycin-induced pulmonary fibrosis rats has suggested the protective effects of NRF2 signaling pathways, including NQO1 and HO1, on oxidative damage, attenuating pulmonary fibrosis (*Ni et al., 2015*). In this study, the regulation of laminarin on these signaling pathways of NRF2 might also imply the potential role of laminarin in treating pulmonary fibrosis via attenuating oxidative damage, inspiring us to explore its application on diseases like pulmonary fibrosis.

In summary, this study identified the protective role of laminarin against hydrogen peroxide-induced oxidative damage in MRC-5 cells, which might be related to the regulation of NRF2 and its signaling pathways. These findings facilitated the understanding of the role of laminarin and the modulation of pulmonary oxidative damage. Further research on more detailed mechanisms would be necessary for the clinical application of laminarin for the alleviation of oxidative damage and the treatment of pulmonary fibrosis.

### Funding
This work was supported by the National Natural Science Foundation of China (No. 81273704), Taishan scholar construction project (No. ts20110819). The funders had no role in study design, data collection and analysis, decision to publish, or preparation of the manuscript.

### Grant Disclosures
The following grant information was disclosed by the authors:
National Natural Science Foundation of China: 81273704.
Taishan scholar construction project: ts20110819.

### Competing Interests
The authors declare there are no competing interests.

### Author Contributions
- Xue Liu and Huaman Liu conceived and designed the experiments, performed the experiments, analyzed the data, contributed reagents/materials/analysis tools, wrote the paper, prepared figures and/or tables, reviewed drafts of the paper.
- Yi Zhai and Yan Li performed the experiments, analyzed the data, reviewed drafts of the paper.
- Xue Zhu performed the experiments, analyzed the data, prepared figures and/or tables, reviewed drafts of the paper.
- Wei Zhang conceived and designed the experiments, wrote the paper, reviewed drafts of the paper.

### Data Availability
The raw data has been supplied as a Supplementary File.

## Supplemental Information

Supplemental information for this article can be found online at http://dx.doi.org/10.7717/peerj.3642#supplemental-information.

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
