# Peer review of "Laminarin protects against hydrogen peroxide-induced oxidative damage in MRC-5 cells possibly via regulating NRF2"

_PeerJ, doi:10.7717/peerj.3642_

## Round 0.1 · original submission · Major Revisions

· Academic Editor

Major Revisions

Two reviewers raise serious questions on the validity of the data. Moreover, from your data it is likely that your cell culture is contaminated with mycoplasma. This should be tested and the results reported in the revised version of the manuscript. If the contamination is confirmed, the experiments should be repeated on a contamination-free cell culture. Appropriate statistical tools should be used and error bars shown where appropriate. Please address the criticism made by reviewers 2 and 3.

Reviewer 1 ·

Basic reporting

1. English is quite acceptable, but still there are some places to be corrected, for example
lien 46 - suggesting
line 62 – myeloperoxidase is not an oxide, it is an enzyme
line 74 – in the lung of bleomycin-induced pulmonary fibrosis
line 219-221 - but the aftertreatment had significant effects when detection was conducted at 24 h post treatment only (P < 0.05).
It seems that prepositions are not always used in correct way, I recommend proofreding by a professional English-speaking editor.
2. Introduction is well written; it gives a good idea about the problem to be solved.
3. Literature well referenced & relevant.
4. The article is well-structured.
5. Figures are of sufficient quality and well-labelled.
6. Raw data are suppled.

Experimental design

1. The manuscript contains results of an original primary research.
2. The research questions are well defined and relevant.
3. The research fills an identified knowledge gap: new knowledge about molecular mechanisms of laminarin action are presented.
4. The manuscript describes results of a rigourous investigation with high technical standart, a vast range of modern research methods have been applied.
5. Methods are described with sufficient detail, moreover standard protocols were mostly used.

Validity of the findings

The article provides a set of data clarifying the role of NRF2 in action of laminarin.

Comments for the author

I think you have performed a nice work, please ask a professional English speaking editor to look through your paper.

Reviewer 2 ·

Basic reporting

no comment

Experimental design

no comment

Validity of the findings

There are doubts about the reliability of some results. Several conclusions are not consistent with the results.
Key remarks:
Fig. 1: Materials and methods indicate that toxicity assessment experiments (MTT) were repeated 6 times. It probably means biological control in one experiment but not independent experiments. The value of OD SD are expected to be much larger in independent experiments as they notably differ in control cells (concentration "0") in Fig. 1A and 1C. This fact should be reflected in Materials and methods.
It is necessary to indicate the SD in Fig. 1A and 1B.
The concentrations of laminarin in Figures 1C and 1D should be indicated in mg /ml and not in relative values.
Fig. 2: there are no SD bars
The authors state that laminarin causes an increase of the protein NRF2 level and promotes its migration into the nucleus that is reflected in Fig. 3B but is not obvious from Fig. 3C.
In Fig. 3C DAPI stains the cytoplasm that indicates the presence of intracellular infection in the culture
Fig.5 A and B - It is shown that laminarin on the background of siNRF2 causes an increase of the level of not only mRNA, but also of the NRF2 protein. Further, in Figure 6A laminarin enhances the expression of NRF2 in cells with overexpression of this protein. How can this be?
As it is shown in Fig. 6C, peroxide and laminarin increase the expression of a number of antioxidant genes against the background of NRF2 being silenced. This finding indicates that the effect of laminarin on the antioxidant genes expression does not depend on NRF2. Nevertheless the authors conclude that the protective effect of laminarin depends on NRF2. Moreover, this is the one of main conclusion of the work.
Fig. 5C: there are no SD bars
The histograms in Figures 5 and 6 use some alphabetic designations (probably statistical significance), which are not deciphered in the signatures.

Reviewer 3 ·

Basic reporting

No comment

Experimental design

No comment

Validity of the findings

The MS by Xue Liu et al. is dedicated to the study of protection function laminarin against pulmonary oxidative damage. The authors also reveal possible mechanisms. I would recommend major revision, because the experiments are not insufficiently detailed and lack appropriate controls.

Major remarks:
1. As the authors investigate the protection function laminarin against oxidative damage, they should include well known antioxidants in this study as positive controls. Laminarin is a polysaccharide and there is possibility that this drug can act not only as antioxidant.
2. Figure 3B contradicts the results obtained in Figure 3C. In Figure 3A and 3B the authors show that laminarin increases mRNA and protein levels of NRF2, but it is not confirmed by immunofluorescence staining (Figure 3C). Also staining with DAPI in Figure 3C shows some contamination, possibly with mycoplasm. This should be tested.
3. Figure 6A. It is unclear how laminarin enhances the expression of NRF2 in MRC-5 cells overexpressing NRF2. This should be explained in detail.
Minor remarks:

1. The authors should indicate the concentrations of reagents in the Figure legends.
2. The manuscript should be edited by a native English-speaker.
3. The list of references should be written in one style. There are errors and inconsistencies on lines 399, 429, 436
4. The authors should refer to original works rather than to reviews in the Introduction and Discussion sections.

---

## Round 0.2 · Major Revisions

· Academic Editor

Major Revisions

Please follow the comments of Reviewer 3 and carry out the following experiment:

1. Comparison of laminarin with the bona fide antioxidants.
2. The Western blot and immunofluorescence experiments are contradictory. Please repeat the experiments and provide convincing data.

Reviewer 1 ·

Basic reporting

The text wa snot edited properly, the changes even worsen it. Here are some my suggestions.


96-97 Further, it has been reported that laminarin had the function of anti-oxidation (Zhou et al. 2009). The specific mechanism was not clear.
Sounds awkward, can be replaced by

It has been reported that laminarin acted as antioxidant (Zhou et al. 2009), although mchanism of this effect was naot understood.

174-175
The biological repeat was performed six times (n=6).

Biological experiments were repeated six times.

212
basing – on the basis of
364
The specific type of interaction needed further explore.

The specific type of interaction should be explored further.
516 and other figure legends
in comparison between Group control and Group H2O2

Control group vs H2O2 Group

Experimental design

All right.

Validity of the findings

Valid.

Comments for the author

Edit English once more, with a native speaker.

Reviewer 3 ·

Basic reporting

No comment

Experimental design

No comment

Validity of the findings

I would recommend major revision, because the experiments are not insufficiently detailed and lack appropriate controls.

Comments for the author

1. The papers cited by the authors (ZHOU et al., Journal of Guangdong Pharmaceutical College 2009 25:397-400; ZHANG et al., Chinese Traditional and Herbal Drugs 2003 34:824-826) can be found nether in the Pubmed not in the internet. The experiments with the well known antioxidants should be included in this study to prove that laminarin acts as an antioxidant.

2. Figure 3C does not show any effects and contradicts the results of immunoblotting. Please repeat the immunofluorescence experiments and provide a convincing image.

---

## Round 0.3 · Minor Revisions

· Academic Editor

Minor Revisions

Please carefully follow the remaining suggestions of the Reviewer.

Reviewer 3 ·

Basic reporting

No comment

Experimental design

No comment

Validity of the findings

I would recommend minor revision.

Comments for the author

Major points:

1. A negative control should be includied in the experiments with Vitamin C in Figure S1.
2. Figure 3C does not show any effects of laminarin under H2O2 treatment and contradicts the results of immunoblotting. Please provide a convincing image.

---

## Round 0.4 · accepted · Accept

· Academic Editor

Accept

The paper is now acceptable for publication.

Reviewer 3 ·

Basic reporting

No comment

Experimental design

No comment

Validity of the findings

No comment

Comments for the author

No comment